# Primary Cilia Influence Progenitor Function during Cortical Development

**DOI:** 10.3390/cells11182895

**Published:** 2022-09-16

**Authors:** Donia Zaidi, Kaviya Chinnappa, Fiona Francis

**Affiliations:** 1INSERM UMR-S 1270, F-75005 Paris, France; 2Sorbonne University, F-75005 Paris, France; 3Institut du Fer à Moulin, F-75005 Paris, France

**Keywords:** primary cilia, cortical development, cortical malformations, neuronal progenitors

## Abstract

Corticogenesis is an intricate process controlled temporally and spatially by many intrinsic and extrinsic factors. Alterations during this important process can lead to severe cortical malformations. Apical neuronal progenitors are essential cells able to self-amplify and also generate basal progenitors and/or neurons. Apical radial glia (aRG) are neuronal progenitors with a unique morphology. They have a long basal process acting as a support for neuronal migration to the cortical plate and a short apical process directed towards the ventricle from which protrudes a primary cilium. This antenna-like structure allows aRG to sense cues from the embryonic cerebrospinal fluid (eCSF) helping to maintain cell shape and to influence several key functions of aRG such as proliferation and differentiation. Centrosomes, major microtubule organising centres, are crucial for cilia formation. In this review, we focus on how primary cilia influence aRG function during cortical development and pathologies which may arise due to defects in this structure. Reporting and cataloguing a number of ciliary mutant models, we discuss the importance of primary cilia for aRG function and cortical development.

## 1. Introduction

The cerebral cortex represents the most superficial region of the brain and is essential for higher cognitive functions, such as thinking and perception [1]. Cortical development in mammals involves a set of highly complex and organised events such as proliferation, differentiation, migration and synaptogenesis. Different cell types essential for corticogenesis are neural progenitors and migrating neurons. 

In mouse corticogenesis, at the onset of neurogenesis, neuroepithelial cells (NECs) give rise to apical radial glial cells (aRG), which are the major neuronal progenitors during cortical development. These are able to self-renew (often via symmetric/horizontal divisions) and also give birth to neurons either in a direct manner or via basal progenitors such as intermediate progenitors (IPs) (generally via asymmetric/oblique or vertical divisions) (Figure 1, [2]). Their unique bipolar shape is composed of a long basal process crossing the cortical wall to reach the pial surface and a short apical process facing the ventricle (Figure 1). aRG soma are restricted to the ventricular zone (VZ), where their nuclei undergo interkinetic nuclear migration, a nuclear movement synchronised with cell cycle advancement: the nucleus moves basally for S-phase, then apically to undergo mitosis at the ventricular surface [3]. A primary cilium (PC) protrudes within the ventricle in interphase and its assembly/ disassembly is linked to the cell cycle (Figure 1) [4,5]. As an antenna-like structure, the PC, a non-motile form of cilium, is composed of 9 + 0 microtubule (MT) doublets. The PC is enriched in receptors to sense the embryonic cerebrospinal fluid (eCSF) and regulates signalling such as Sonic hedgehog (Shh), Hippo, Notch and Wnt [6], thus aiding the control by extracellular cues of aRG proliferation and scaffolding [7]. 

PC formation relies on other organelles such as the centrosome. Centrosomes are major microtubule organising centres (MTOC) formed by 9 MT triplets and surrounded by a complex of proteins termed pericentriolar material. They are composed of a pair of centrioles: the daughter centriole (younger) and the mother centriole (older) that exhibit distal (DA) and sub-distal (sDA) appendages allowing MT anchorage. When centrosomes move towards the plasma membrane (apical membrane for aRG) in G0/G1, PC formation starts after the mother centriole transforms into a basal body. The extracellular pathway involves the docking of the mother centriole to the plasma membrane followed by axoneme elongation. The intracellular pathway relies on the fusion of the mother centriole with ciliary vesicles and secondary vesicles allowing axoneme extension already within vesicles before arriving at the plasma membrane [8,9,10].

aRG play a critical role in cortical development such as maintaining the neuronal progenitor pool, producing neurons and acting as a support for neuronal migration. When alterations in aRG function occur, this can lead to cortical malformations. These are a set of pathologies that are generally visible with magnetic resonance imaging. We describe in this review, PC alterations leading to a variety of malformations (Figure 2). Defects can lead to micro- (too small) and macro-cephaly (too large) ([11,12,13,14] and references therein), affecting the maintenance of the progenitor pool and neurogenesis. We discuss below the potential origins of such disorders. Certain PC defects in aRG can also be associated with heterotopia (neurons in the wrong position). Thus, certain forms of periventricular (PH) and subcortical heterotopia (SH) can be associated with mutations in genes influencing the PC (Figure 2) affecting progenitor position and integrity. Polymicrogyria (an excess of folds on the surface of the brain) can also be linked with these disorders [11,13,14,15].

The association of ciliary/centrosomal protein mutations and cortical malformations, confirms the crucial role of PC in cortical development. Excellent reviews explain how cilia are assembled and disassembled [16,17] but, here, we review how PC specifically impact aRG behaviour and their progeny in normal cortical development and disease. In Section 2, we present upstream mechanisms of PC formation such as centrosome dynamics and cytoskeleton remodelling, followed by a description of the cell cycle and its link to the PC. Then, in Section 3, we explain how PC dynamics impact aRG proliferation, differentiation and neuronal migration, discussing mouse and human mutations. Finally, in the last part (Section 4) we discuss key emerging points regarding ciliary mutant mouse phenotypes, signalling, cilia size and centrosomes.

## 2. Upstream Mechanisms of Primary Cilia Formation

### 2.1. Initiation of PC Formation and Centrosome Dynamics

The PC, an antenna-like structure emanating from the apical surface of the basal body, is based on an axoneme structure consisting of a radial arrangement of nine MT doublets lacking a central pair (9 + 0 structure) [18]. Transition fibres, below the transition zone (TZ), anchor the axoneme to the ciliary membrane at the base of the PC. They help in compartmentalising this organelle from the cytosol and regulate protein entry and exit into the PC [19]. Protein transport into the PC, including the transport of tubulin subunits to replace tubulin that is disassembled from the axoneme tip, is supported by an intraflagellar transport (IFT) system, powered by plus (+) and minus (−) end MT motor proteins, kinesin and dynein respectively, [20] (Figure 3A). The PC membrane is enriched with several transmembrane receptors and signalling phosphoinositides that are important for sensing and signalling functions [6]. Membrane cargos, to concentrate and regulate signalling molecules in the ciliary membrane, are trafficked in vesicles by the Bardet-Biedl Syndrome (BBSome) coat complex [21].

The series of cycles of cilia assembly and disassembly are coupled with changes in cytoskeletal dynamics involving mainly actin and MTs [22,23,24,25]. The centrioles play a major role in acting as a template for the growth of axoneme MT structure. PC assembly and disassembly are highly coordinated with the cell cycle in actively proliferating cells, with the PC being present in G0/G1 when the centrosome is at the apical pole and disappearing in S/G2 when the centrosome is required for mitotic spindle formation prior to mitosis [16]. The factors influencing the movement of the centrosome to the apical side and the cytoskeletal changes assisting this process remain an understudied topic. MTs have been shown to regulate the positioning of the centrosome at the cell centre/cell surface by exerting pushing and pulling forces [22,26]. Apart from this, the actin cytoskeleton is also known to regulate centrosome positioning, with actomyosin contractility assisting basal body migration to the apical pole [23,24]. To understand specifically the involvement of the cytoskeleton in the migration of the centrosome towards the apical surface during primary ciliogenesis, Pitaval and colleagues [22] carried out detailed live imaging studies in RPE1 cells. Deletion of selected well-known ciliogenesis factors including Cep164, Cep123, Ift20, Pard3, nesprin2, meckelin and Kif3a was found to affect centrosome positioning at the apical pole also leading to defective PC formation. On the other hand, Ift88 and pericentrin were found to be solely important for axoneme elongation. 3D tracking in cells specifically deficient for the DA protein Cep164 showed severe defective migration of the centrosome towards the apical pole, validating further that the defective positioning is caused by abnormal centrosomal migration rather than just an anchoring problem at a later stage. Investigation of the MT involvement in this process showed that increased MT nucleation/polymerisation and transient stabilisation lead to densification and bundling of the MT network that exerts pushing forces to propel the centrosome towards the apical pole. In addition, simultaneous symmetry breaking of radial actin arrangement leading to preferential clustering of actin on one side, driven by myosin II activity, was observed along with the asymmetric co-partitioning of MTs (Figure 3B), providing direct evidence for the coordination of actomyosin contractility and MT stabilisation driving centrosome migration prior to PC formation. It is also interesting to underline the disturbance in cytoskeletal remodelling processes upon Cep164 deletion, further suggesting coordination between ciliogenesis factors and the cytoskeleton in these processes [22].

### 2.2. Other Cytoskeletal Processes Contributing to PC Growth

The cytoskeletal network involving both actin and MTs is known to play a coordinated role and impact PC formation by assisting the series of steps right from the migration of the basal body to the apical pole as described above, its docking with the plasma membrane and then delivery of ciliary membrane and cargo to a growing PC [27,28,29].

Taxol-induced polymerisation and stabilisation of cytoplasmic MTs in RPE1 cells are known to destabilise and shorten PC as a result of reduced levels of the soluble pool of tubulin, while nocodazole-mediated depolymerisation of cytoplasmic MTs leads to increased PC number and length, confirming the necessity for the maintenance of sufficient levels of cytoplasmic soluble tubulin to support ciliary growth and maintenance [30]. Further, it is presumed that the MT-sDA connections facilitate the transport of PC-targeted vesicles towards the ciliary base, although this may require further confirmatory studies [29,31].

Likewise, actin polymerisation is in general known to negatively regulate the ciliogenesis process causing thus PC disassembly or inhibition of ciliogenesis, while depolymerisation is associated with ciliary assembly and elongation [25]. However, several studies in different cell types and model systems on the contrary also showed that the disruption of the actin cytoskeleton leads to defects in PC formation and maintenance, suggesting the importance of the actin network for basal body migration, apical docking and regulation of the entry of ciliary cargo, including the Ift particles at the ciliary base [32,33]. The actin cytoskeleton also helps with the transport of vesicular cargo powered by myosin motors myosin II and myosin Va to deliver membrane in the form of vesicles, Ift components and other cargo during ciliogenesis, clearly suggesting the importance of proper actin cables to facilitate this transport [28]. Thus, induced depolymerisation of actin is known to arrest vesicle transport, with the degree of depolymerisation differentially affecting ciliogenesis. Also, several actin-related proteins are sometimes important for regulating cilia formation, by controlling YAP/TAZ activity [28,30,34,35]. All these indications together suggest a complex involvement of actin dynamics in time and space to regulate ciliogenesis processes. 

### 2.3. PC and Cell Cycle

aRG actively go through the cell cycle whilst still attached to the ventricular surface. From other dividing cells, it is known that cilia dynamics are intimately linked with cell cycle and proliferation [36,37,38]. PC expression can be repressed in proliferating cells, such as in cancer cells, thus allowing faster advancement of the cell cycle and faster proliferation [5,39,40]. However, how one process influences the other remains not fully understood. We know that in certain mammalian cells, PC deciliation occurs in several waves that coincide with cell cycle re-entry and cell cycle checkpoints [5,25,41], however recent papers also suggest that in other mammalian cells, entire PC loss can occur in one go [42]. Nevertheless, generally, cells start to disassemble their cilia coinciding with cell cycle re-entry, then they reassemble it. When defects in these first disassembly steps are observed, cell cycle re-entry delays occur [43]. Second waves of deciliation occur to allow entry of cells in the S-phase, contributing to the G1/S phase transition [25,44,45]. Later, generally prior to mitosis in late G2, the PC can be fully resorbed [25]. For the G1/S transition, trichoplein activates Aurora A to inhibit cilia formation and triggers deciliation. When there is a knockdown of these proteins, cells are stuck in the G1 phase suggesting that they are essential for S-phase entry [46]. Notably, Tctex-1, a dynein light chain, is phosphorylated and recruited at the base of the PC to promote ciliary disassembly and S-phase entry in ciliated cells but is dispensable in non-ciliated ones [47]. Similar observations are true for the centrosomal protein Nde1: when depleted, cells exhibit longer cilia and retarded G1/S transition [43]. These data strongly suggest that cilia disassembly and key related factors are essential to allow the correct G1/S transition, as well as cell cycle advancement (see discussion in Section 3 for additional proteins regulating these steps).

## 3. The Impact of Primary Cilia on Neuronal Progenitor Behaviour during Cortical Development

### 3.1. PC Assembly versus Disassembly for Proper Proliferation

#### 3.1.1. Cilia Formation Defects and Impacted aRG Proliferation 

It has long been thought that the PC must be totally disassembled prior to mitosis [48,49], but Paridaen and colleagues showed in aRG that a shortened cilium remains (ciliary remnant, CR, Figure 1 and Figure 3) that is internalised with the basal body in G2/M [50]. This CR is asymmetrically inherited in aRG. In their study, they characterised how CR inheritance impacts cilia reformation and aRG behaviour [50]. They showed that when the CR is inherited by a cell, it reforms a cilium faster and thus leads to asymmetric ciliary signalling between the two daughter cells. The CR is often inherited in a daughter cell that maintains stem cell properties in early cortical development. PC formation in aRG is thus linked with proliferative behaviour. Also, the CR is known to be associated with the older mother centriole giving these daughter cells a stem cell character. These findings regarding the CR are coherent with previous works that suggest that the older centriole is preferentially inherited in the aRG that will maintain the stem cell character and complete thus the understanding of centrosomes-CR asymmetric inheritance [51]. This association rate of the CR-older centriole decreases over corticogenesis, corresponding to the neurogenic phases. The impact of the CR on stem cell properties (or differentiation) has been covered in depth in other reviews [15].

One important step during ciliogenesis in polarised cells such as aRG, after centrosome movement, is the anchoring of the mother centriole to the apical plasma membrane, thus allowing ciliary axoneme elongation (see Section 2). A major protein in this phenomenon is Cep83, a DA protein (DAP). A recent study using a Cep83 conditional knockout (cKO) mouse model, exhibiting an enlarged cortex and abnormal folding of the mouse brain, showed that *Cep83* deletion leads to a failure of centrosome anchoring to the apical plasma membrane and PC formation defects [52]. These defects in centrosome anchoring lead to excessive proliferation of aRG. The authors compared their results with an Ift88 cKO mouse model exhibiting PC formation defects but no centrosome anchoring defects, nor cortical defects [52]. They conclude that centrosome anchoring, more so than the PC itself, is the most important factor and raised the question of to what extent cilia are required in cortical development compared to centrosomes. It should be mentioned, however, that the Ift88 mutant mouse model does exhibit cortical malformations when the deletion is performed during other specific spatiotemporal periods [53]: using *Emx1*-Cre expression, the Ift88 mutation does not lead to major defects, but when deletion is induced using *Foxg1*, *Wnt1* and *Ap2*-Cre expression (earlier and/or different locations), cortical defects are clearly observable [53]. These combined results are interesting with respect to the role of Ift88 and PC during corticogenesis.

Mutations in several proteins leading to human cortical malformations (patients suffering from micro- or macro-cephaly, polymicrogyria or heterotopia, see Table 1) have been shown to negatively influence cilia number and/ or length (likely due to centrosome/PC formation defects). We can mention *NINEIN* and *CENPF*, which are implicated in syndromes associated with microcephaly and that both colocalise at the sDAs of the mother centriole [54]. NINEIN is described to be required for the reassembly of the centrosome following mitosis [55]. Also, shortened PC were observed in *CENPF* mutant fibroblasts and kidney cells [54,56,57]. CENPF is also expressed during mitosis and its mutation in Hela cells leads to mitotic delay [58]. *KIF14* mutations can also lead to microcephaly and have been shown to disturb PC formation and disrupt the localisation of the DAPs SCLT1 and FBF1 as well as components of the IFT-B complex [59,60]. KIF14 is known to function during mitosis, its depletion has been described to impact cytokinesis, as well as to increase S- and G2/M-phase cells. Cilia defects, seen during serum starvation, could, however, be dissociated from mitotic defects [60]. SCLT1, and TBC1D32 a TZ protein, are both required for initiating ciliogenesis, with mutations giving rise to severe ciliopathies (OFD type IX) including microcephaly [16,61,62]. *Sclt1* loss in mouse decreases the number of cilia in the kidney and increases proliferation (Ki67+ cells) but also apoptosis of renal tubule epithelial cells [63]. Depletion of Tbc1d32 leads to reduced and dysmorphic cilia since it is important in TZ integrity and ciliary bud formation [16,64].

Guo et al. [72] also delineated specific functions of a series of human ciliopathy-associated genes during different steps of cortical development in the mouse. From this, we can cite further examples of key PC genes playing a role in aRG. These include, for example, *BUBR1*, *IFT80*, *KIF7* and *TMEM216*, important for cortical progenitor cell division [72]. Some mutations in these genes give rise to microcephaly, with also evidence for shortened cilia. An example is the *BUB1B* gene [136], playing a role in the basal body, affecting docking and PC growth [71,73]. BUB1B also plays a role at the spindle-assembly checkpoint and its mutation leads to disrupted mitotic progression, aneuploidy and massive cell death. Loss of Bubr1 in the mouse allowed progenitor cells to prematurely bypass the mitotic checkpoint, thereby accelerating the progression of mitosis and effectively decreasing the proportion of mitotic cells among cycling cells ([73], Table 1). Also, *Rotatin* (*RTTN*) mutations can lead to microcephaly, as well as polymicrogyria or simplified gyral patterns [123,124]. In patient fibroblasts, mutant RTTN was correctly localised to basal bodies and axonemes, but PC were also shortened, and sometimes presented a thickened, bulbous appearance [123]. Knockdown also leads to shorter cilia with interestingly, multiple basal bodies [123]. Centrosome amplification was also observed in patient fibroblasts, perturbing mitosis [124]. Indeed, RTTN also plays a role at the spindle poles and in bipolar spindle formation and its depletion leads to a significant increase in multipolar aneuploid cells, causing cell death. RTTN mutation resulted in fewer cells entering the G1 phase, G2/M cell cycle arrest and a prolonged cell cycle ([124], Table 1). STIL and CEP135 are postulated to be interactors of RTTN. These microcephaly genes when mutated also perturb centrosome number, and lead to disorganised spindles, metaphase arrest and cell death ([125,126], Table 1). Thus, Stil -/- mouse embryonic fibroblasts (MEFs) lacked centrosomes and PC [128]. Patient fibroblasts from *CEP135* patients showed, however, multiple fragmented centrosomes, disorganised MTs, as well as reduced cell growth rate [78]. 

The heterotopia gene *EML1*, coding for a MT-binding protein, shows mutations leading to macrocephaly and polymicrogyria [85,86,87]. Cilia that are reduced in number, as well as shortened, are observed in human and mouse models [87,89]. Mitotic spindles were normal morphology but enlarged [137]. Overall, cell cycle exit was found to be decreased in the mouse model ([85], Figure 4). Also, related to EML1, specific heterotopia mutations in *RPGRIP1L*, a Joubert syndrome gene coding for a protein at the TZ, also lead to a reduced number and length of PC in human fibroblasts [87]. The same PC phenotype is observed for *TCTN2* mutations, a TZ protein mutated in Joubert and Meckel syndromes with some children exhibiting heterotopia [114,131,132]. Furthermore, *AHI1* mutations can also give rise to Joubert syndrome with some patients showing polymicrogyria [114]. This protein is enriched at the base of the PC, associated with the basal body, and mutations disrupt ciliary signalling and reduce proliferation in cerebellar neurons [67]. Patient missense mutation plasmids transfected in non-neuronal cells were reported to perturb ciliary formation [68], as was gene knockdown [67]. However, no ciliogenesis defects were observed in mutant cerebellar cells, despite disrupted ciliary signalling [67]. *OFD1* mutations are associated with focal malformations, or Joubert syndrome with some children exhibiting heterotopia, macrocephaly and polymicrogyria [113]. Under normal conditions, OFD1 has been described to inhibit ciliogenesis, which is promoted when OFD1 is degraded at centriolar satellites [115]. Inversely, its accumulation at centriolar satellites leads to fewer and shorter PC [115]. Possibly paradoxically, OFD1 mutant protein has been shown to disrupt cilia formation, potentially because of changed autophagy at the cilia [116]. Mutant fibroblasts were also shown to have disrupted cell cycle progression [117].

FLNA, a classical PH protein [90], was shown to localise to the basal body with meckelin. Fewer patient cells showed cilia, they had reduced length and basal bodies were also shown to be mispositioned [93]. In a separate study, mouse mutants for FlnA were shown to have reduced progenitors with a prolonged cell cycle, leading to a reduced brain size [92]. Of note, however, a majority of PH patients do not have changed brain size [90]. Also, *CSPP1* mutations give rise to Joubert syndrome, with or without Jeune asphyxiating thoracic dystrophy. Three out of ten patients were also shown to exhibit PH [84]. CSPP1 localises to centrioles as well as spindle poles, and human patient fibroblasts similarly show reduced numbers and/or short PC. RNAi depletion of *CSPP1* also perturbed cell-cycle progression in the late S-phase [83].

Because the PC is intricately linked with centrosomes at basal bodies, and many proteins localise to this region, it can be difficult to determine the most important and primary event leading to ciliogenesis abnormalities (e.g., centrosomes). Many proteins are also expressed during mitosis, giving rise to phenotypes potentially independent of the PC. Similarly, we cannot exclude in all cases, that mutations do not impact the disassembly process which could also contribute to PC formation defects. We now discuss genes and proteins, which have, however, been more clearly described to influence PC disassembly. 

#### 3.1.2. Numerous and/or Elongated Cilia and Impaired aRG Proliferation 

Protruding within the ventricle, the PC senses eCSF cues. Recently, lysophosphatidic acid (LPA), a bioactive phospholipid, was identified to play an important function in aRG proliferation via its role in cilium disassembly. LPA signalling thus modulates Aurora A to trigger cilia disassembly [138]. This also involves YAP/TAZ, actors of the Hippo signalling pathway. LPA via promotion of cilium disassembly hence promotes proliferation.

Also, Cenpj (Cpap) is known to act as a scaffold for the cilium disassembly complex, which includes Nde1 and Ofd1 (mentioned above), as well as Aurora A, allowing tightly regulated disassembly [81]. *In vitro*, neuronal progenitor cells (NPCs) derived from microcephaly-patient induced pluripotent stem cells (iPS) with mutations in *CENPJ*, were found to exhibit longer cilia suggesting PC disassembly defects. Patient progenitors presented delayed cell cycle re-entry leading to premature differentiation [81]. These PC results were confirmed in aRG *in vivo* in a CenpJ cKO mouse model exhibiting microcephaly [82]. The disassembly defect was associated with reduced aRG proliferation and unfinished mitosis [82]. Kif2a overexpression rescued this defect suggesting that this protein via its MT depolymerising activity is also involved in cilium disassembly [82,139]. In human, *KIF2A* mutations give rise to a complex cortical disorder involving microcephaly as well as cortical dysplasia [98]. In human fibroblasts and mouse models, elongated cilia are observed and there is delayed progression through mitosis [99]. NDE1, for which mutations give rise to microcephaly and lissencephaly [109,110], is normally degraded upon entry into G1 (like OFD1 mentioned above). Enhanced Nde1 leads to a reduction in cilia length and depleted Nde1 in lengthening [43]. Patient NDE1 proteins are unstable, cannot bind cytoplasmic dynein, and do not localise properly to the centrosome [109]. Mutant cells also show multipolar spindles and M phase arrest [109].

In the same way, in *WDR62*/*Wdr62* mutant cerebral organoids and a KO mouse model, during cortical development progenitors exhibit cilium disassembly defects associated with decreased proliferation and increased differentiation contributing to microcephaly ([80], Table 1). Indeed, Wdr62 has been shown to recruit Cep170 to the basal body, this protein interacting with Kif2a to drive cilium disassembly. Overexpression of KIF2A also rescued PC disassembly defects as well as a proliferation defect in *WDR62* mutant NPCs. Wdr62 also has been shown to recruit Cenpj to the basal body for cilium disassembly [135] suggesting that Wdr62 is the upstream protein in this cilium disassembly pathway (Figure 4). Interestingly, another group studying the role of a *WDR62* human variant in mouse [134] showed microcephaly and depletion of the aRG pool, but it was associated with less ciliated cells and smaller cilia suggesting increased disassembly, or PC formation defects. The reasons for such differences in cilia length between these studies are unclear but they could be due to the fact that the studied WDR62 human variants (WDR62^R439H/R439H^ and WDR62^V66M/V66M^) in mice lead, respectively, to maintained or reduced expression of WDR62 but not the total loss of the protein [134], as is the case for the KO mouse and KO organoids [80]. However, also Wdr62^stop/stop^ mice exhibited a drastic reduction in Wdr62 expression and shorter cilia as well and one explanation for such differences could be the genetic background of studied mice as suggested by authors [134]. However, in both cases PC are defective, leading to a microcephaly phenotype, strongly suggesting that loss of function of Wdr62 is important for the aRG pool. It is clear that it is important to take into account genetic backgrounds and actual patient variations to fully understand all phenotypes. 

Further human genes associated with forms of microcephaly are, for example, *KIF11*, for which mutations have been shown to lead to elongated cilia. Little is known about the role of this protein in cortical neural progenitors; however, it is expressed during mitosis and was shown to associate with the daughter centriole during interphase [105,106]. *KATNB1* mutations also leads to microcephaly and lissencephaly, with null fibroblasts showing an excess of centrioles and cilia, as well as mitotic spindle defects [96]. In the Drosophila optic lobe, asymmetric division of neural progenitors was reduced with Katnb1 loss. There is also a delayed anaphase onset of central brain neuroblasts and reduced cell number, independent of cell death [96].

This catalogue of mutations and phenotypes may collectively suggest that when the PC is longer and more numerous (most often due to disassembly defects), this can impair cell cycle progression and lead in mouse and human to microcephaly (Table 1). Of note, however, as we saw above in Section 3.1.1, microcephaly can also be associated with shortened cilia, depending on the mutant gene. Thus, there are clearly different outcomes leading to microcephaly, even grouping those mutations leading to elongated cilia, since mutations can lead to premature neurogenesis and thus depletion of the progenitor pool, or mitotic arrest and cell death, depending on the varied functions of microcephaly genes.

### 3.2. aRG Differentiation

#### 3.2.1. Neurogenesis and Basal Progenitors 

aRG can form neurons in a direct or an indirect way, the latter via basal progenitors such as IPs. The role of the PC in contributing to this decision between forming neurons directly or indirectly remains to be clearly elucidated. Nevertheless, we provide, in this section, several examples of how PC impact aRG neurogenesis and differentiation (Figure 4). 

Inpp5e is an enzyme that localises to the ciliary membrane [94,140] and controls the inositol phosphate composition of the latter. *INPP5E* is also a Joubert syndrome gene [94,140]. Hasenpusch-Theil and colleagues showed that during mouse cortical development when Inpp5e function is lost, aRG PC are abnormally formed. In the lateral neocortex at early corticogenesis, decreased IP number and an increased number of neurons suggest that direct neurogenesis occurs preferentially, however, this is not the case in the medial neocortex [95]. IP generation is no longer affected at mid-corticogenesis in the lateral neocortex but is altered in the medial cortex. There is hence a spatio-temporal aspect of the phenotype since direct neurogenesis is first affected in the lateral, and later in the medial neocortex [95]. The Shh Gli3 repressor (Gli3R) is reduced in these mutants and re-introducing Gli3R rescues the decreased formation of basal progenitors [95] suggesting that Inpp5e regulates neurogenesis via Gli3 signalling. It is interesting to note that in a Tctn2 mutant (related to a gene coding for a protein important for TZ architecture, and also involved in Joubert syndrome and heterotopia as mentioned above [130,141]), aRG also preferentially underwent direct neurogenesis early in development [95]. The comparisons of these two ciliary mutants hence showed that cilia integrity is essential for the proper balance in neurogenesis and particularly at early stages. 

Another Kif protein that is essential for PC dynamics is Kif3a. Indeed, Kif3a knockdown experiments using *in utero* electroporation [102] lead to ciliogenesis alterations as well as several defects in aRG functions, especially proliferation by delaying cell cycle progression, but also neuronal migration. Mimicking Shh signalling with Gli2 overexpression rescued aRG cell cycle lengthening via an increased expression of cyclin D1. 

In Wdr62 mutants or double Wdr62, Aspm mutants, in early cortical development, CR are found prematurely dissociated from centrosomes and are correlated at mid-corticogenesis with the increased generation of IPs. The authors proposed that this is due to the loss of the mother centriole with which CR can hence no longer associate [135]. PC defects are hence secondary in this case to the centriole phenotype. Centriole-CR association is thus important for the maintenance of stem cell character and defects in these processes could lead to premature differentiation into IPs as discussed above.

Also, the localisation of the PC during division is related to the generation of basal progenitors as showed by Wilsch-Bräuninger and colleagues [142]. Basolateral localisation of the cilia in nascent cells is a rare event in early corticogenesis when high aRG proliferation is required, but it increases with the progression of neurogenesis. Indeed, basolateral cilia occurrence is higher (around 50%) when the generation of basal progenitors such as IPs occurs. Insm1 overexpression (involved in the production of basal progenitors) in mouse leads to an increased proportion of basolateral cilia rather than apical [142]. Because apical cilia sense cues from eCSF, we can imagine that basolateral cilia will lead to a different sensing in the nascent cell, thus suggesting a role of cilia localisation in aRG progeny production.

#### 3.2.2. Neuronal Delamination 

Neurons and basal progenitors produced from aRG in the VZ need to detach to migrate to their proper locations in the cortical wall. This phenomenon is known as delamination. As well as IP production, aRG delamination (via detachment and/or oblique division, in [143]) can give rise to basal RGs (bRGs), especially in gyrencephalic species [143]). aRG detachment can also be pathological [52,85] leading to ectopic aRG. Very little is known about the involvement of cilia in pathological delamination even though several studies associate PC defects and ectopic aRG or premature or aberrant production of IPs ([11,87,89,144], see also below). In this section, we focus on PC behaviour related to aRG progeny delamination (e.g., neurons and basal progenitors).

During neurogenesis, apical abscission can occur to allow neuronal delamination. During these processes, PC are dismantled as it has been shown in chick spinal cord using live imaging [145]. Notably, centrosomes are conserved by the detaching cells whereas the cilium remains attached to the abscised apical membrane. The authors of this work suggest that separation of centrosomes and cilia during abscission may reduce active Shh signalling, as they observed an accumulation of Smo and Gli2 in cells about to perform abscission. Similarly in developing mouse cortex, in line with the above studies, the association of CR with the centrosome is found to decrease later, compared to earlier stages, when differentiation and delamination of cells are increased [50]. Further, a pre-delamination event with the appearance of basolateral cilia on a proportion of polarised neuroepithelial cells (NECs/aRG), has been observed to give rise to basal progenitors. The proportion of these cells increases over the course of neurogenesis, with increasing delamination events giving rise to differentiated progenitors and neurons [17]. Also, the centrosomal protein Akna is known to promote delamination. Associating specifically with the delaminating/differentiating apical progenitors and basal progenitors, it was found upon downregulation to cause no differences in the proportion of short/long cilia on apical neural stem cells, however it would be interesting to look also into the formation of basolateral cilia in these cells [146].

#### 3.2.3. Neuronal Migration

It is very likely that the PC also influences subsequent neuronal migration and the formation of proper neuronal circuits. As mentioned above, in an attempt to understand the importance of several ciliopathy genes across cortical development, Guo and colleagues [72] utilised a shRNA approach to independently target several ciliopathy genes via *in utero* electroporation in mouse embryos and performed detailed analyses to study the consequences across development. In addition to several genes identified to influence the apico-basal polarity and division of aRG, neuronal differentiation and connectivity, a majority of these genes (17 out of the 30 studied) were identified to impact neuronal migration and the final placement of these cells in the cortex. The delay in neuronal migration caused upon the knockdown of these genes was found to be due to either defects in the multipolar to bipolar transition of neurons, affecting also the size and number of processes of the multipolar neurons, or increased branching of the leading process, which could also be shortened in length, increased length of the trailing process, as well as misorientation of the bipolar neurons. These results clearly indicate that ciliopathy gene mutations impact all aspects of cortical neuron migration including the transient multipolar stage, multipolar-to-bipolar transition and RG-guided radial migration of neurons [72].

mTOR hyper-activation identified in the models of focal malformations of cortical development was found to lead to defective ciliogenesis linked to inhibition of autophagy and accumulation of the autophagy substrate OFD1 [115,147,148]. Park and colleagues identified perturbed Wnt signalling as a cause for polarised migration defects observed upon mTOR hyperactivation and OFD1 accumulation, and this is possibly mediated through the PC [148]. It would be further interesting to study the possible impact of defective PC in the process of multipolar to bipolar transitioning, as mentioned above. Also, it is worth noting that mutations in genes known to cause neuronal migration disorders such as *FLNA* (as well as *LIS1* and *DCX*) may interfere in this multipolar to bipolar transition process in young migrating neurons [149].

It is known that both the radially migrating cortical projection neurons and tangentially migrating interneurons display PC. However, conditional deletion of the small GTPase Arl13b (a classical PC marker) in either migrating cortical neurons or interneurons using *Nex*-Cre/ *Dlx5/6*-Cre was found to cause no defects in the radial migration of cortical projection neurons, while however, disrupting severely the tangential migration of interneurons [150]. Arl13b mutant migrating interneurons are found to undergo fewer translocation events with the periods of pausing characterised by the neurons sequentially extending processes in multiple directions without any accompanying somal translocation events, the overall reduction of the migration rate compared to the control and failing to sense the guidance cues [150]. Thus, the PC is clearly important in this cell type. Although it has been argued that the PC would be more important for the extracellular non-substrate-based guidance of interneuron migration, rather than the substrate-based radial migration of cortical neurons, the importance of cilia in the multipolar to bipolar transitioning of young neurons before they acquire directionality for migration, could be better analysed in the future by deleting *Arl13b* earlier, for example, by utilising *Tbr2*-Cre. Moreover, *Nex*-Cre conditional deletion of *Arl13b* in cortical neurons did not seem to disrupt the structure of the PC, therefore suggesting that the importance of these organelles in radial migration of cortical neurons still cannot be completely negated, although the relative importance for tangential migration of interneurons could still be higher [150]. 

In organisms with folded cortices, such as in the case of the human brain, the cortical projection neurons display both radial and tangential modes of migration compared to the stricter radial migration mode in rodents [151]. Thus, time-lapse imaging studies utilising a gyrencephalic model, the ferret, identified lateral dispersion of these neurons migrating along several different radial fibres [152]. Compared to the phenotypes mentioned above, it is further interesting that relatively more extensive branching of leading processes of migrating neurons was identified in ferret compared to mouse and that the wide-angle branching particularly identified in ferret preceded the sharp turns to change directions to favour tangential migration of these neurons [153]. Therefore, increased branching in gyrencephalic species across evolution is proposed to favour the tangential migration of cortical neurons and cortical folding. Yanagida and colleagues [154] showed that branch selection in interneurons depends on the movement of the Golgi/centrosome in the chosen branch, which may also suggest an involvement of the PC. Notably as well, mouse interneurons also show increased branching compared to radially migrating neurons [155]. Given these reasons, it is possible that PC may have a differential impact between mouse and human cortical neuron migration, therefore it could be important in the future to study the effects of PC defects in human cortical migration disorders utilising appropriate models.

### 3.3. aRG Morphology

#### 3.3.1. Polarity 

aRG are characterised by their unique polarised morphology. In interphase, they exhibit a bipolar shape with a short apical process facing the ventricle from which protrudes the PC, and a long basal process crossing the cortical wall to reach the pial surface (Figure 1). It is interesting to note that some ciliary proteins have been linked to aRG scaffolding and shape. We can mention again the GTPase Arl13b, shown in mutant mice carrying a null allele, termed *Arl13b^hennin(hnn)^* [69,156], to exhibit drastically perturbed apico-basal polarity of aRG, with an inverted polarity (Figure 4). This suggests that Arl13b may be essential for the initial formation of the aRG scaffold. It is likely to modulate signalling essential for neuronal progenitor polarity. 

Afadin is an F-actin binding protein important for cell-cell junctions, involved in maintaining epithelium integrity, such as the neuroepithelium. It has been shown that loss of Afadin in early cortical development leads to PC abnormalities in aRG. PC were shorter and less numerous and there was a redistribution of ciliated neuronal progenitors from their correct positions in the VZ. We can note that the first abnormalities that were observed in *Afadin* inactivation mutants were PC defects at embryonic day 12.5 (E12.5), followed by a loss of polarity proteins at E13.5. Thus, Afadin is important for proper PC formation influencing progenitor positioning [157]. Guo et al. also identified BBS1, 7 and 10, as playing a role in aRG polarity, after knockdown experiments in the mouse [72].

#### 3.3.2. Scaffold and Localisation

Eml1 shows mutations in an SH spontaneous mouse model [85], as well as in rare atypical heterotopia patients (Table 1). Many aRG were delaminated and those in the VZ exhibited shorter and less numerous cilia [87], defects observed even from E12.5 in the mouse, suggesting an essential role of Eml1 in PC formation. As well as aRG delamination, an abnormal aRG scaffold was observed [85,137] and there was abnormally increased proliferation. In human cerebral organoids deficient for EML1, aRG cilia defects were also observed. Increased YAP1 expression was shown to be involved in aRG mispositioning [89] since its inhibition with verteporferin and fluvastatin leads to a significant decrease of basally positioned aRG. Of note, we have also observed increased nuclear YAP in the Eml1 mouse mutant (unpublished data). YAP is a downstream effector of the Hippo signalling pathway which can be regulated by the PC [158]. When YAP is activated (nuclear translocation), it increases the expression of PC disassembly factors such as Aurora A and Plk1 [159,160]. Inversely, when YAP is cytoplasmic, this is correlated with increased cilia formation.

aRG apical endfeet terminate in a membrane domain that is essential to form a barrier with eCSF in the ventricle. The integrity of the apical domain can influence proliferation, mechano-transduction and attachment/delamination (e.g., producing basal progenitors and neurons) [161]. During corticogenesis, aRG apical domains increase in size, leading to changes in overall apical membrane stiffness [162]. Ciliary abrogation in early corticogenesis (E11) in a Kif3A cKO mouse model, also leads to an enlargement of the aRG apical domain, as shown by *en face* imaging from E12.5 [144]. This phenotype is linked with increased production of Tbr2+ basal progenitors [144]. The enlargement in the apical domain, leading to decreased ventricular wall integrity, is thought to be due to cilia signalling defects, and rapamycin treatment (mTorc1 inhibitor) rescues aRG enlargement and ventriculomegaly. Similarly, in the Eml1 spontaneous mutant mouse (HeCo), it has been also shown that apical domains are enlarged compared to WT [87,137]. YAP signalling in the VZ is also mechano-sensitive, and Shao et al. [52] showed that centrosome anchoring to the apical membrane controls the mechanical properties of aRG, a process intimately linked with the PC.

These studies suggest that cilia (probably via their signalling activity) are important for correct aRG morphology and positioning.

## 4. Key Remarks

### 4.1. Spatio-Temporal Cilia Deletion Matters

The timing of gene deletion, when studying cilia function during mouse cortical development, seems to be crucial, as well as the spatial aspects of the deletion. As mentioned in our review, the study of several mouse mutants exhibiting cortical and PC defects reveals the complexity of determining the specific impact of cilia on cortical development and aRG.

Indeed, *Ift88* deletion does lead to PC defects but not systematically in aRG nor always leading to cortical defects [52,53]. For example, conditional deletion of *Ift88* at early stages (E9.5) via *Foxg1*-Cre expression leads to E14 and E18 cortical defects, but deletion of *Ift88* using *Emx1*-Cre expression leads to fewer defects at E14 and no more damage at E18. Similar observations are true for the Kif3a mutant mouse [53].

Furthermore, when deletion of Arl13b is performed at E9.5 (using *Foxg1*-Cre), E10.5 (*Nestin*-Cre) and E13.5 (h*GFAP*-Cre), it does not result in the same phenotype. When cilia are depleted early (potentially in NECs or early aRG), aRG scaffolding is abnormal with perturbed apico-basal polarity (Figure 4), but this does not occur at latter stages [69].

It should be noted in general that PC defects are likely to show a cell-type specificity, most probably related to levels of gene/ protein expression. It is hence sometimes difficult to extrapolate to cortical cells when analyses are initially performed in other cell types. The models mentioned in Table 1 allowing studies of neural progenitors are hence important to be able to relate defects and their consequences to cortical development.

We can imagine that NECs or early aRG are more sensitive to signalling via PC at early stages. We can highlight Hippo signalling, which is linked with mechano-transduction signals since the stiffness of the ventricular wall changes over corticogenesis [162]. Also, CR and centrosomes are less frequently associated as cortical development proceeds [50]. Dynamics and requirements of PC, CR and other structures change over time, so it is reasonable to conclude that cilia ablation will not lead to the same consequences when deleted early or late.

Furthermore, in the Wdr62 mutant condition, it has been shown that cilia can be either longer in aRG or absent with premature dissociation of the CR from the mother centrosome [80,134,135]. Even though cilia defects are not the same, interestingly, aRG differentiation to basal progenitors was increased in both cases. Thus, with a different cilium defect, the aRG phenotype can in some cases, at least regarding differentiation/proliferation, appear the same.

Transgenes/Cre drivers used for conditional deletion are thus important and reveal that cilia defects impact cortical development with respect to a specific spatiotemporal pattern that still needs to be fully elucidated and may help explain divergent results obtained in several studies.

### 4.2. Signalling via the PC Is Crucial

We highlight here the importance of ciliary signalling on aRG behaviour. Through their signalling role important for cell cycle, proliferation, autophagy, etc., cilia influence key functions in aRG. Signalling through the PC in neuronal progenitors has already been well reviewed [158,163] and here we focus on rescue experiments performed by ciliary signalling modulation.

First, via mTorc1 inhibition (with rapamycin), Foerster and colleagues showed that it is possible to rescue aRG apical domain enlargement in the ciliary mutant mouse Kif3a cKO, as well as the ventriculomegaly [144]. This strongly suggests that via mTorc1 signalling, PC play a role in ventricle morphology as well as aRG apical endfeet integrity.

Recently, studying EML1 mutant organoids, treatment with verteporferin and fluvastatin (Hippo modulators) was shown to partially rescue the phenotype of detached aRG, which is the major phenomenon leading to SH formation, described in Section 3 [85,89]. PC defects observed in Eml1 mutant aRG seem to be thus more than a coincidence and most probably directly linked to aRG detachment. However, we should mention that YAP/TAZ can be activated through different mechanisms and cannot be solely and always related to PC.

When *Pard3* is deleted early in mice (E9.5), this leads to an enlarged cortex with SH. aRG were altered in terms of their number and organisation. Although cilia were not verified in this study, the authors showed that dual removal of the Hippo pathway effectors YAP and TAZ suppresses cortical defects observed in Pard3 lacking mice [164], suggesting the possibility of some impacts caused by PC signalling defects, although this remains to be exactly shown.

In the Inpp5e mutant, the PC and the cortex were defective, and it is of interest to note that Shh Gli3 repressor (Gli3R) ratio restoration rescued such defects. Indeed, the PC is involved in the regulation of full-length Gli3 (Gli3FL) cleavage into Gli3R, via TZ proteins such as Tcnt2 and Rpgrip1L. Thus, PC integrity and Shh signalling are crucial for correct aRG and cortical development [95].

Several rescue experiments involving modulation of ciliary signalling thus indicate the probable importance of the PC, via its signal transducer role, in aRG during corticogenesis.

### 4.3. Cilia Size Matters?

It is of interest to note that in ciliary mutant mice, some aRG may retain a cilium even if its size is affected. Related to this, cilia size may be an important criterion for its function. Indeed, in some mutant mice, PC length is affected, giving rise to aRG phenotypes, even though cells are theoretically still ciliated. Mechanisms controlling PC length have already been well reviewed by Keeling et colleagues [165]. We mention here several mutant examples to further discuss cilia length versus function.

Gli2 KO NIH3T3 fibroblasts possess a longer cilium after serum starvation due to increased autophagy, dependant on Ofd1 degradation as shown by Hsiao and colleagues [166]. Because of longer cilia, cell cycle re-entry is delayed in the absence of Gli2 [167]. Similarly, cells depleted for Nde1 (a mother centriole protein) in zebrafish embryos exhibited a longer cilium and a delayed cell cycle re-entry that was correlated with ciliary length [43]. Thus, cilia size is important for such processes as suggested by authors. We know that PC, centrosomes and cell cycle are intimately linked and axoneme length regulates cell cycle time [5,167]. We can propose that longer cilia may lead to delayed cell cycle re-entry because of a longer cell cycle.

Also, Tuberous Sclerosis Complex proteins 1 and 2 (TSC1 and TSC2) regulate cilia length in mouse embryonic fibroblasts (MEFs), impacting PC function such as Shh signalling [168]. However, we cannot exclude the fact that even if cilia size is affected, signalling may not be totally impaired. Indeed, cilia can be shorter but still receive some signals. For example, in the Eml1 mutant mouse condition, aRG cilia were shorter, but there were no obvious defects in Shh signalling [87]. However, single-cell analyses are required to completely confirm this result, and indeed other signalling pathways appear perturbed (e.g., Hippo signalling, and unpublished data).

### 4.4. Cilia Abnormalities, Causes or Consequences of Cortical Defects?

PC dynamics are tightly regulated and involve multiple events and factors presented in the above sections. It is still unclear whether PC could be the sole factor leading to such cortical and aRG phenotypes. Different opinions on the question emerge, suggesting either pure cilia, centrosome or both phenotypes [17,52].

It is also of interest to note that ciliary proteins could play a role outside the cilia and impact the interpretation of phenotypes in ciliary mutant mice, as already shown for the small GTPase Arl13b [169]. Indeed, Arl13b plays a role independently from the PC, in mediating neuronal axon guidance in the spinal cord [169], and we can hence potentially imagine a role outside the PC in aRG as well. However, it is of interest that the studies by Higginbotham and colleagues uncovered the role of Arl13b specific to the PC during neuronal migration, upon its deletion, and by attempting rescue studies with a non-ciliary form of Arl13b that retains the GTPase activity but not the PC function [150]. Also, Ift88 plays a role outside the cilia in MCDK cell migration as shown by Bohlke and colleagues [170]. Furthermore, Ift88 has been shown to be involved in the G1/S transition in non-ciliated cells [171]. Non-ciliary roles hence need to be taken into account.

Because PC formation relies on centrosomes, it is legitimate to raise the question regarding cilia defects (hence aRG and cortical defects) as causes or consequences of centrosome defects. PC keep centrosomes in interphase and then PC resorption allows centrosomes to be released for mitosis, thus in this way, it is not easy to distinguish between the cell cycle, the centrosome and PC-specific roles. Another confounding factor is that many basal body proteins are also expressed during mitosis. This is particularly the case for a number of previously mentioned microcephaly proteins, many of whose centrosomal levels are regulated in a cell cycle-dependent manner (e.g., [75], see cilia formation section). Thus, proliferation abnormalities may in these cases be separate from cilia defects. It remains unclear if these cortical malformations should be viewed as ciliopathies or should be refined as centriolopathies. Indeed, it is hard to imagine that if centrosomes are defective, cilia will be normal, but the contrary could be possible, proper centrosomes can be linked to defective PC signalling and structure as mentioned above. It is however important to also note that in RPE1 cells, several ciliogenesis factors were shown to control the positioning of the basal body at the apical pole prior to ciliogenesis, possibly affecting the migration of the basal body by regulating actomyosin contraction and MT stabilisation, clearly interlinking the two processes [22].

*Cep83* deletion known to eliminate DAs of the mother centriole, affects centrosomal apical membrane anchorage leading to activated YAP and excessive aRG proliferation with excessive production of neurons and an enlarged cortex (discussed above, [52]). *Ift88* deletion leading to proper anchorage to the apical membrane with intact DAs caused no dramatic defects as compared to *Cep83* deletion, suggesting that the centrosome is more relevant for the observed phenotype than cilia. However, studies in RPE1 cells showed that Ift88 and pericentrin were solely important for axoneme elongation but not centrosomal migration to the apical pole which was regulated by several different ciliogenesis factors as explained in Section 2.1 [22]. It is known that certain factors related to Hippo signalling, including some localised to the basal body of the centrosome, are important for proper ciliogenesis. Disruption of their organisation/function could therefore correlate with dysregulated Hippo signalling (reviewed by Wheway and colleagues, [158]), in addition to the cell-extrinsic mechanism acting through LPA and PC regulating the same pathway [138]. All these data suggest the importance of the interlink between proper basal body formation and the associated ciliogenesis process. Therefore, the extent of defects in the entire process might correlate with the severity of phenotypes observed, and the importance of PC cannot be neglected, that is, considering that these multiple steps are highly interlinked.

Also, it has not yet been systematically studied if centrosomes are defective in many of the ciliary mutant mouse models described. A systematic verification could thus be really informative to know if PC defects (hence aRG and cortical defects) are always correlated with centrosome impairment or not. Live imaging would be useful to help determine the extent of the defects because it should provide information on these dynamic processes.

The complexity and the vast interplay between the PC, centrosomes, cell cycle and other subcellular processes are still unclear, but the recent advances in the field as well as the great interest in cilia/centrosome studies in neuronal progenitors may give us further ways to help dissect these mechanisms and their impact on cortical development and aRG in the future. Spatial centrosome proteome studies carried out recently by O’Neill and colleagues [172] comparing the interactome changes of different centrosomal proteins at different locations between neural progenitors and neurons could also help identify the specific roles of different proteins and their interactions in the associated processes, as well as specific functions in different cell types, while also extending this information to understand the pathologies caused by mutations in these different factors [172].

## 5. Conclusions

In this review, we have mainly focused on the role of the PC in neuronal progenitors. Many studies provide evidence on why PC are key organelles in cortical development, and notably impact aRG behaviour. Concerning the centrosome, a cell cycle-dependent organelle, it is very hard to attribute the sole role of cilia to the mutant phenotypes observed. However, emerging omics datasets and cilia protein mutations in several cortical malformations do indeed emphasise the importance of the PC. They have been shown to impact aRG proliferation, differentiation and scaffolding during cortical development. Briefly, we can conclude that in aRG, problems in PC formation may be associated with increased proliferation of neuronal progenitors and thus sometimes macrocephaly. Inversely, disassembly defects could lead to aRG premature differentiation and/or cell death and thus often microcephaly. However, as mentioned in Section 3, there are many variants of this simplified view. Many factors and multiple pathways are for sure involved in proliferation and differentiation, which are the main functions of aRG, and many proteins can influence PC machinery (including centrosomal proteins). Future advances in the field will certainly continue to identify and characterise precise mechanisms by which the PC specifically influences aRG behaviour during corticogenesis.

## Figures and Tables

**Figure 1 cells-11-02895-f001:**
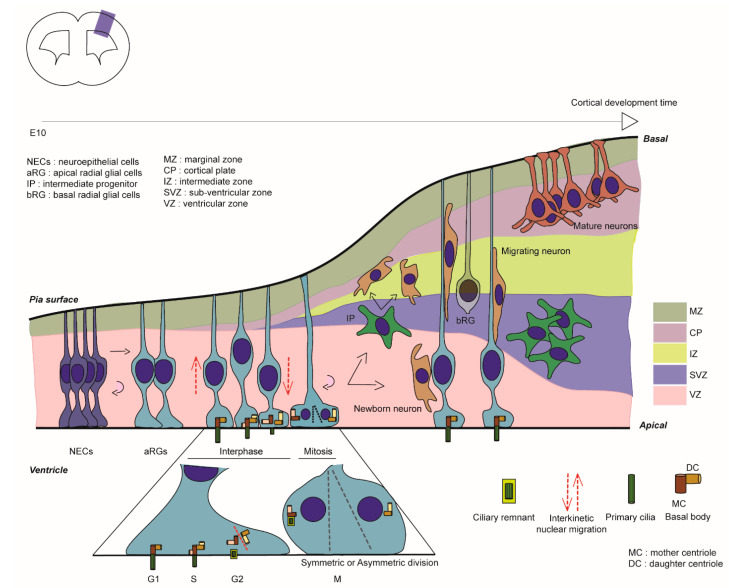
Cortical development in mouse. Graphical representation of mouse cortical development. Colour codes have been used to represent the different regions in the developing cortical wall. Enlargement represents a simplified representation of centrosome/primary cilia dynamics related to cell cycle advancement in aRG. Cortical development involves different steps and cell types that are tightly orchestrated. aRG are able to self-amplify, but also give birth to other cell types such as neurons (post-mitotic) and basal progenitors (IPs, bRGs) that will in turn give birth to neurons.

**Figure 2 cells-11-02895-f002:**
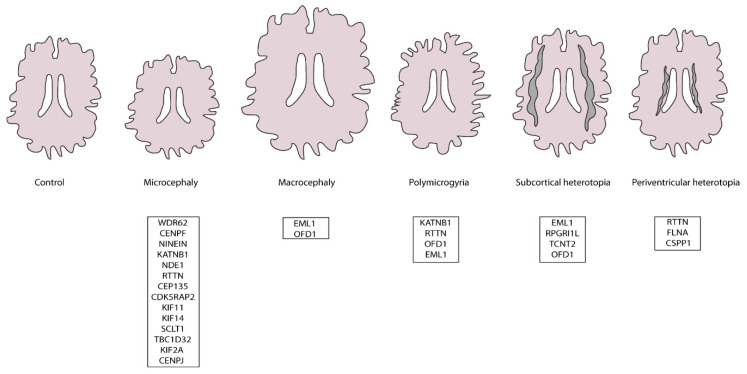
Schematic representation of human cortical malformations mentioned in this review and related ciliary genes that are known to be mutated in human patients.

**Figure 3 cells-11-02895-f003:**
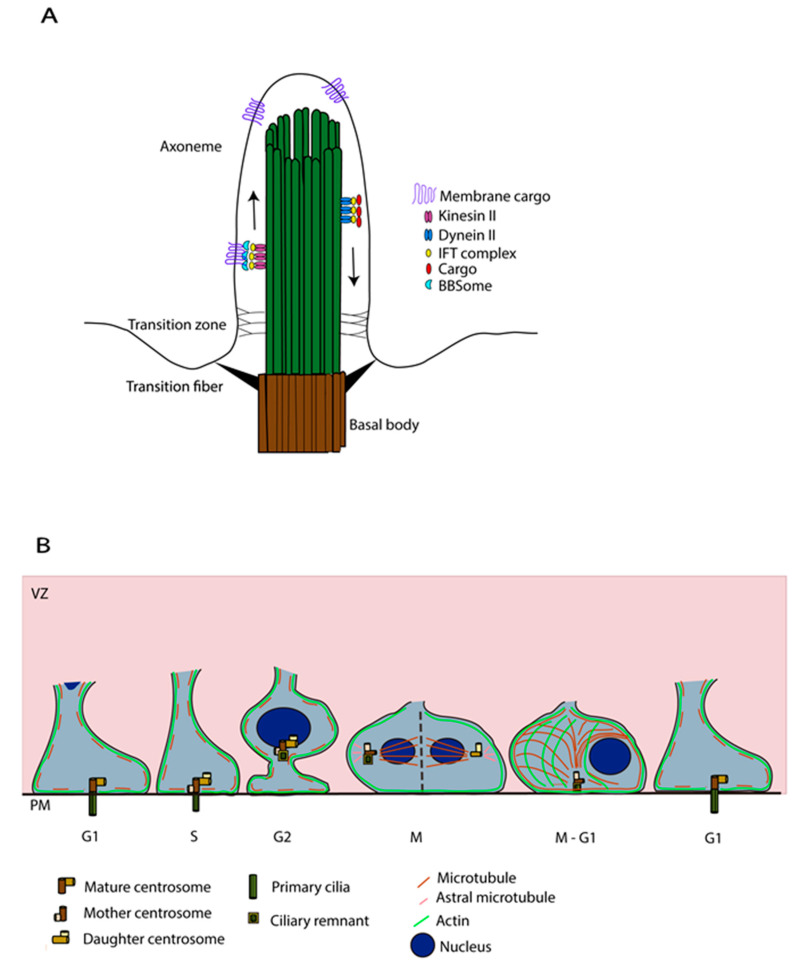
Upstream mechanisms of PC formation in aRG. (**A**) Schema showing the structure of PC, based on 9 + 0 array of MT doublets (green), anchored at the basal body (brown). Kinesin II motor proteins associated with the IFT complex support the anterograde transport of cargos, while dynein II motor proteins associated with IFT complex help with the retrograde transport of cargos. Association of IFT motors with the BBSome complex supports the transport of membrane cargos. The transition zone at the base of the PC helps block diffusion of cytosolic and membrane proteins. (**B**) Dynamics of PC and centrosomes in the VZ of the developing neocortex across the cell cycle. The PC is assembled across G1 interphase and starts going through the process of disassembly during S phase, before entering the Mitotic (M) phase when it is completely disassembled, or at times still carries the ciliary remnant (which can help establish the PC sooner in one of the progeny). During this PC assembly process, specific actomyosin contraction and MT stabilisation help drive the basal body to the apical pole to dock and initiate PC assembly (this schema related to cytoskeletal dynamics is adapted for neural progenitors based on the studies carried out in RPE1 cells [22]). PM. Plasma Membrane, VZ: Ventricular Zone.

**Figure 4 cells-11-02895-f004:**
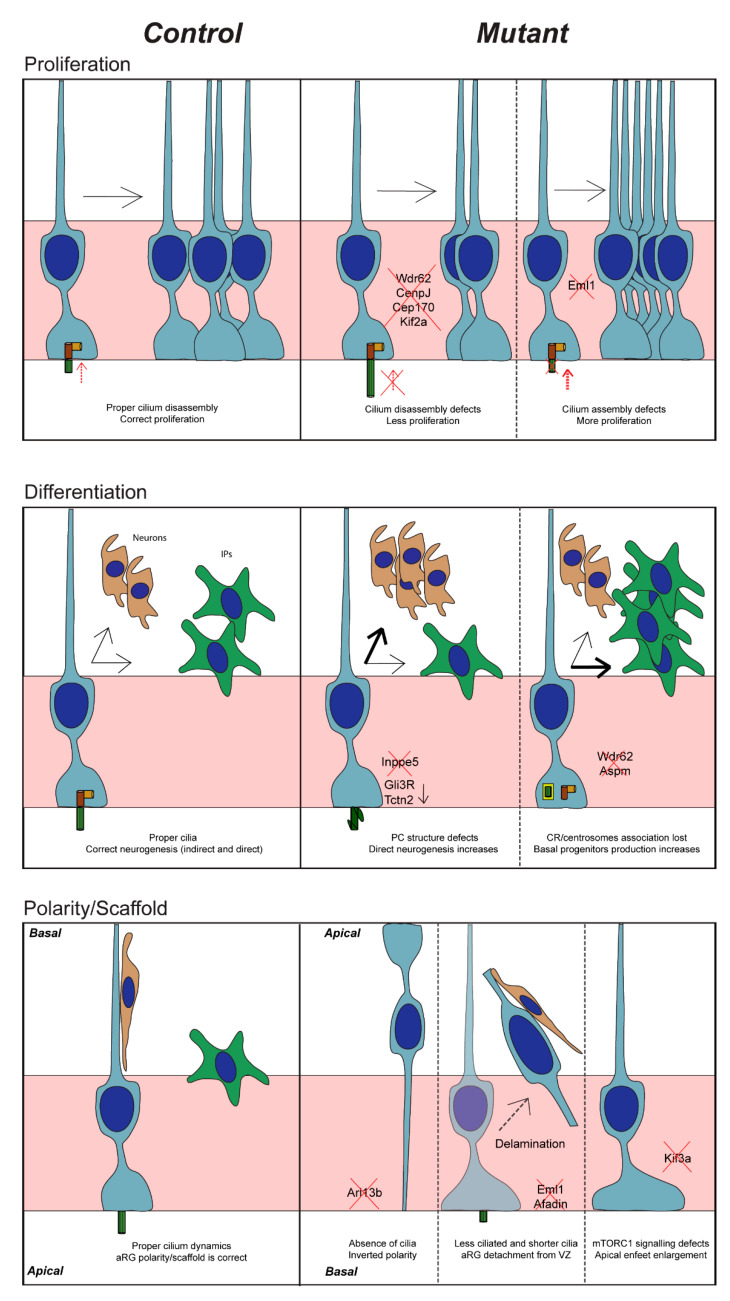
PC impacts on aRG behaviour. Graphical schema of PC defects and their impact on aRG. On the left, the control situation for proliferation, differentiation, scaffold/polarity are represented and in the right panel, the mutant condition with a non-exhaustive list of involved ciliary factors. The red cross refers to an absence of proteins or an alteration of a process.

**Table 1 cells-11-02895-t001:** Ciliary genes and proteins mutant in mouse and human in cortical malformations. Table representing a non-exhaustive list of genes involved in PC formation and function found mutated in human (blue-grey) and mouse (green). Genes are listed in alphabetic order. AR, autosomal recessive. AD, autosomal dominant.

Gene	Function/Localisation	Phenotype	Chromosome	Brain Malformation	Models	Mutant	Phenotype	References
*AHI1*	Enriched at interface of cilium and basal body (mother centriole).	Some reductions in cilia formation observed with AHI1 patient missense mutation by transfection.	6 (AR)	Joubert syndrome, polymicrogyria.	Zebrafish and mouse models Human constructs were transfected into IMCD3 cells. Human fibroblasts also studied.	KO mouse (full)	Hypoplastic cerebellum with an underdeveloped vermis and mildly defective foliation pattern.	[65,66,67,68]
*ARL13B*	Small GTPase localised at the cilium membrane.				Mouse	Arl13b null	Absence of cilia. Disrupted aRG scaffold and polarity. Neuronal heterotopias.	[69]
*BUB1B/BUBR1*	Localised at basal bodies. Involved in proteasomal degradation. Also required for mitotic progression.	Abnormal docking of centrosomes, fewer cells with cilia.	15 (AR)	Mosaic variegated aneuploidy (MVA), including severe intrauterine growth defect and microcephaly.	Medakafish and mouse models Human fibroblasts.	Hypomorphic Bubr1 allele (Bubr1 H/H mice) Bubr1 cKO (*Emx1*-Cre).	Defect in metaphase plate formation and shortened metaphase. Reduced cortical ventricular surface and cortical progenitors, including apical and intermediate progenitors, massive cell death in progenitors and cortical neurons.	[70,71,72,73]
*CDK5RAP2*	Centrosome cohesion and engagement, thereby restricting centriole replication.	Centriole amplification with a preponderance of single, unpaired centrioles and increased numbers of daughter-daughter centriole pairs. Multipolar spindles during mitosis, also excess of mother centrioles leading to multiple primary cilia.	9 (AR)	Microcephaly.	Mouse	KO mouse (full) CRISPR/Cas9-mediated genetic KO.	Centriole amplification with many single, unpaired centrioles and increased numbers of daughter-daughter centriole pairs. Delayed chromosome segregation and chromosomal instability in neural progenitors.	[74,75,76]
*CENPF*	Localises to the basal body in ciliated fibroblasts and at the subdistal appendages of the mother centriole.	Shortened cilia.	1 (AR)	Stromme syndrome, microcephaly.	Zebrafish and mouse models Human renal epithelial cells and fibroblasts.	KO mouse (full)	Loss of ciliary structure, tubule dilation, and disruption of the glomerulus in the kidney.	[54,56,57,77]
*CEP135*	Present at the centrosomes, in pericentriolar material and during the early phase of centriole elongation. Critical function in early centriole and basal body assembly. Regulation of centriole duplication.	Spindles are disorganised, cells either lack centrosomes and cilia, or have multiple fragmented centrosomes.	4 (AR)	Microcephaly.	Patient fibroblasts Zebrafish and mouse models	CRISPR/Cas9-mediated genetic KO.	Centriole duplication defects, TP53 activation, and cell death of progenitors. Trp53 ablation prevents cell death but not microcephaly, and it leads to subcortical heterotopias.	[76,78]
*CEP164*	Mother centrosome protein. Important for anchoring of centrosomes to the plasma membrane to initiate ciliogenesis.				Mouse	Cep164 cKO (*Foxg1*-Cre)	Hydrocephalus.	[79]
*CEP170*	Centrosomal protein located at the subdistal appendage of the mother centriole. Known to promote cilium disassembly.				Human NPCs and Mouse	shRNA mouse	Longer cilia.	[80]
*CPAP/* *CENPJ*	Centriole duplication. Scaffold for cilia disassembly complex.	Elongated cilia with delayed cell cycle re-entry. Premature neurogenesis and cell death increased.	13 (AR)	Seckel syndrome, microcephaly.	Mouse and cerebral organoids	CenpJ cKO (Emx1-Cre)	Longer cilia. Decreased proliferation and increased differentiation. Microcephaly.	[74,81,82]
*CSPP1*	Centrosome and spindle pole associated protein 1.	Fibroblasts from affected individuals with CSPP1 mutations showed reduced numbers of primary cilia and/or short primary cilia. RNAi depletion leads to perturbed cell cycle progression.	8 (AR)	Joubert syndrome with or without Jeune asphyxiating thoracic dystrophy. 3/10 patients with heterotopia (periventricular).	Human fibroblasts and Zebrafish			[83,84]
*EML1*	MT-binding protein found in the vicinity of centrosomes.	Heterotopia mutations lead to reduced primary cilia length and number. Cell cycle exit is decreased.	14 (AR)	Heterotopia (subcortical), macrocephaly, polymicrogyria, corpus callosum agenesis.	Human fibroblasts and Mouse	Human organoids Spontaneous (HeCo) & mouse KO (full)	Shorter cilia. aRG detachment. Decreased cell cycle. Subcortical heterotopia.	[85,86,87,88,89]
*FLNA*	Basal body localisation with meckelin.	Knockdown leads to ciliogenesis defects. Fewer patient cells show cilia. Cilia have reduced length. Basal body mispositioning.	X-linked	Heterotopia (periventricular).	Patient fibroblasts (c.1587delG p.K529fs) and FlnaDilp2 null. Zebrafish and mouse models	KO mouse (full)	Death at midgestation with widespread hemorrhage from abnormal vessels.	[90,91,92,93]
*IFT88*	Intraflagellar transport protein. Localised at centriole and cilium basal body. Involved in the anterograde transport of ciliary protein.				Mouse	Ift88 cKO (*Foxg1*, *Ap2*-Cre)	Bigger brain and dysmorphic.	[53]
*INPP5E*	Enzyme located in the ciliary membrane which hydrolyses the phosphatidylinositol polyphosphates PI(4,5)P2 and PI(3,4,5)P3.				Mouse	Inpp5e Δ/Δ	Defective cilium membrane and structure. Transient increase in direct neurogenesis. Thinner cortex.	[94,95]
*KATNB1*	Regulates overall centriole, mother centriole, and cilia number.	Katnb1 null fibroblasts show an excess of centrioles and cilia. Defective mitotic spindle formation. Affects asymmetric division leading to reduced cell number.	16(AR)	Complex cerebral malformations (including microcephaly and lissencephaly spectrum, polymicrogyria).	Multiple mutant models: Drosophila, Zebrafish, Human cells, Mouse	KO mouse (full)	Increased cell death. Reduced cycling and proliferating radial neuroepithelial progenitor.	[96,97]
*KIF2A*	MT depolymerisation protein involved in cilia disassembly.	Delayed progression through mitosis. Cilia disassembly defects and elongated cilia.	5 (AD)	Microcephaly, cortical dysplasia, complex, with other brain malformations (- 3), (agyria, posterior predominant pachygyria, subcortical band heterotopia, and thin corpus callosum).	Human fibroblasts (e.g., KIF2A p. His321Asp), and Mouse	KIF2A (p. His321Asp) conditional knock-in mouse	Increased cell death. Smaller brain and neuronal cortical layering defects.	[80,98,99,100,101]
*KIF3A*	MT motor protein localised at the centriole and cytoskeleton. Involved in MT anchoring at the mother centriole.				Mouse	sh*Kif3a* (IUE)	Less ciliated neural cells. Delayed cell cycle progression. Bigger brain.	[53,102]
*KIF11*	MT motors. Associates with daughter centriole.	Increased number and increased length of primary cilia in mutant cells.	10 (AD)	Microcephaly with or without chorioretinopathy, lymphedema, or mental retardation.	Multiple cell types Human RPE1 cells (Crispr/Cas9 to generate heterozygous state).			[103,104,105,106]
*KIF* *14*	MT motors. Located in the cilia in interphase. Also plays a role during mitosis.	Kif14 absence hampers the efficiency of primary cilium formation and the dynamics of primary cilium elongation. Disrupts the localisation of the distal appendage proteins SCLT1 and FBF1 and components of the IFT-B complex. Also deregulates Aurora A activity.	1 (AR)	Microcephaly or Meckel Syndrome.	Multiple cell types and Mouse Human hTERT RPE-1 cell line used for siRNA studies.	Spontaneous (homozygous splice site mutation in the *Kif14* gene caused loss of the wildtype protein)	Smaller brain with dysgenesis of the cerebral and cerebellar cortices and the hippocampus. Severe hypomyelination of the brain and spinal cord. Massive neuronal cell death.	[59,60,107,108]
*NDE1*	Component of ciliary disassembly complex.	Ciliary defects linking disassembly and cell cycle progression. Enhanced Nde1 leads to reduction in cilia length. Depleted Nde1 leads to lengthening and delayed cell cycle re-entry.	16 (AR)	Lissencephaly with microcephaly. Microhydranencephaly.	Patient lymphoblasts Zebrafish and Mouse models	KO mouse (full)	Small-brain phenotype. In MEFs, defects in mitotic progression and increased mitotic index.	[43,109,110]
*NINEIN*	Part of the MT-anchoring complex proteins. Ninein is enriched at subdistal appendages.	Centriole maturation defects.	14 (AR)	Seckel syndrome: severe short stature, microcephaly, and developmental delay.	Hela cells and Zebrafish Human fibroblasts.			[51,55,111]
*OFD1*	OFD1 is expressed during mitosis. In interphase it is localised at the base of the cilium..	Disruption of ciliogenesis and mitotic arrest also observed in mutant fibroblasts. Autophagic degradation of OFD1 at centriolar satellites promotes primary cilium biogenesis. When OFD1 accumulated at centriolar satellites, fewer and shorter primary cilia are observed.	X-linked (XLR and XLD)	Joubert syndrome, but some children can also have heterotopia (subcortical), macrocephaly, polymicrogyria, corpus callosum agenesis and also focal malformations.	Zebrafish and mouse models	KO mouse (leads to an aberrant mRNA encoding a truncated protein of 106 aa)	Dorso-ventral patterning defects. Absence of ciliary axonemes but presence of mature basal bodies correctly orientated and docked.	[112,113,114,115,116,117,118]
*PCM1*	Pericentriolar material protein. Important for centrosomes assembly and function.				Mouse	Pcm1 +/− mouse	Reduced cortex volume.	[119]
*RPGRIP1L*	Located at the transition zone. Involved in the degradation of proteins via proteasome.	Heterotopia mutations lead to reduced primary cilia length and number, reduced interaction with EML1.	16 (AR)	Joubert and Meckel syndromes. Children can have heterotopia (subcortical), and corpus callosum agenesis.	Human fibroblasts and Mouse	KO mouse (full)	Shorter cilia. Defect in symmetric vs. asymmetric division. Thinner cortex.	[87,120,121]
*RTTN*	Co-localises with basal bodies and axonemes. Determines early embryonic axial rotation, as well as anteroposterior and dorsoventral patterning in the mouse.	In fibroblasts, abnormally short cilia with multiple basal bodies. Centrosome amplification, cell cycle arrest (G2/M) aneuploidy.	18 (AR)	Microcephaly, short stature, heterotopia (periventricular) and polymicrogyria with seizures. Simplified gyral pattern.	Human fibroblasts and Mouse	KO mouse (full)	Embryonic lethality, with deficient axial rotation, notochord degeneration, abnormal differentiation of the neural tube. Loss of the left-right specification of the heart. Severe hydrocephalus.	[122,123,124]
*SCLT1*	SCLT1 is recruited to centrioles distal appendage. Important for docking and initiating ciliogenesis.	Complete cilia loss.	4 (AR)	OFD Type IX, including microcephaly.	Zebrafish and mouse models	KO mouse (full)	Decreased number of cilia in kidney. Increased proliferation and apoptosis of renal tubule epithelial cells.	[61,62,63]
*STIL*	Centriolar assembly protein.		1 (AR)	Microcephaly	Mouse	KO mouse (full)	Reduced size and limited developmental progress. Midline neural tube defects, including delay or failure of neural tube closure and holoprosencephaly.	[125,126,127,128]
*TBC1D32*	Transition zone protein required for ciliogenesis.	Reduced and dysmorphic cilia.	6 (AR)	OFD Type IX, including microcephaly.	Zebrafish and mouse models	Mouse with recessive splice-site mutation (bromi=Tbc1d32)	Exencephaly with absence of cephalic ventral midline furrow, poorly developed eyes, and preaxial polydactyly. Neuroepithelium with curled axonemes surrounded by dilated ciliary membranes	[62,64,129]
*TCTN2*	Localised at the transition zone. Required for transportation of proteins into the cilia.	Significant reduction of the number of cilia in the neural tube. Basal bodies docked normally to the plasma membrane. Hedgehog signalling disrupted.	12 (AR)	Joubert and Meckel syndromes. Children can have heterotopia (subcortical).	Mouse	KO mouse (full)	Lacked nodal cilia. Cilia in neural tubes are scarce with defective morphology and failed to elongate axonemes.	[114,130,131,132]
*WDR62*	Involved in the regulation of spindle assembly and orientation. Plays a role in centrosome inheritance, and maintenance.	Retarded cilium disassembly, long cilium, and delayed cell cycle progression leading to decreased proliferation and premature differentiation of NPCs. Defects may lead to G1-S phase delay.	19 (AR)	Microcephaly, pachygyria.	Mouse and cerebral organoids	KO mouse (full)	Longer cilia. Decreased proliferation and increased differentiation. Microcephaly.	[80,133,134,135]

## Data Availability

No data are provided in this review.

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
