# Peer review of "Primary Cilia Influence Progenitor Function during Cortical Development"

_cells, 2022, doi:10.3390/cells11182895_

Round 1

Reviewer 1 Report

The processes of corticogenesis are tightly controlled by a series of cellular events, including neural progenitor proliferation, differentiation and positioning of newborn neurons. Accumulating lines of evidence suggest roles of primary cilium (PC) in the radial glia, the primary cortical neural progenitors, in regulating cortex development and implications in related brain defects. In this manuscript, the authors comprehensively summarize the progresses in PC formation, function and cortical defects related with cilia abnormalities. In general, this manuscript should be of potentially interests for researchers in the field of cortex development and related diseases. I have only a few points that could be considered during revision of the manuscript.

1.      The manuscript has listed many references and some of them might not be closely related with the main theme. Please consider only retain those most relevant references.

2.      Journal names are repeated in all the references. Please check and correct.

3.      Figure 3. The middle “G1” should be “M”.

4.      Table 1. Function/Phenotype should be described more concisely and coherently.

5.      Many sentences are vague and should be rephrased, such as lines 283-284, 296-298, 312-313, 323-324, 381-382, 560-563, 588-589, 648-649, etc.

6.      The contents of some paragraphs like lines 375-380 are fragmented and incoherent with the context. Please check the whole manuscript.

Author Response

The processes of corticogenesis are tightly controlled by a series of cellular events, including neural progenitor proliferation, differentiation and positioning of newborn neurons. Accumulating lines of evidence suggest roles of primary cilium (PC) in the radial glia, the primary cortical neural progenitors, in regulating cortex development and implications in related brain defects. In this manuscript, the authors comprehensively summarize the progresses in PC formation, function and cortical defects related with cilia abnormalities. In general, this manuscript should be of potentially interests for researchers in the field of cortex development and related diseases. I have only a few points that could be considered during revision of the manuscript.

We would like to thank the reviewer 1 for the time spent on the revisions of our manuscript. We are very happy to read that our work was appreciated. We have taken in account all the recommendations to improve the manuscript including the suggestions for English/grammar/style corrections that we hope will improve the reading of the manuscript.

  1. The manuscript has listed many references and some of them might not be closely related with the main theme. Please consider only retain those most relevant references.

We were careful during the revisions to check and only cite the references that are the most closely related to the theme and we removed a number that were less closely related.

  1. Journal names are repeated in all the references. Please check and correct.

We apologize for these errors in the original reference list. We have re-done the list and now paid attention to the correct editing of all references. 

  1. Figure 3. The middle “G1” should be “M”.

We thank the reviewer for noticing this error. We have changed the figure 3 as mentioned. In addition, we corrected some small typographical errors in Figures 2 and 4.

  1. Table 1. Function/Phenotype should be described more concisely and coherently.

We edited the table as suggested and homogenized the function/phenotype columns in the new Table which we feel now has an improved presentation. 

  1. Many sentences are vague and should be rephrased, such as lines 283-284, 296-298, 312-313, 323-324, 381-382, 560-563, 588-589, 648-649, etc.

Thank you for helping us identify sentences to improve. As well as those indicated we have closely proof-read the entire manuscript to improve the writing and style.

  1. The contents of some paragraphs like lines 375-380 are fragmented and incoherent with the context. Please check the whole manuscript.

Incoherent content of some paragraphs, long and confusing sentences were re-written and simplified, and we hope this has improved the quality of the manuscript.

Reviewer 2 Report

In this review the authors provide an extensive overview of the role of primary cilia (PC), protrusions that form on the apical process of radial glia during early corticogenesis. In addition to centrosomes, they are associated with many important functions of apical radial glia allowing the proper cortical development to occur. The author provide a detailed description of the PC formation, structure and functions as well as a "catalogue" of ciliary genes/proteins linked to the brain malformations when mutated (listed in Table 1 with brief description of their function, both in wild type and mutated phenotype and with references for the specific studies performed). The rest of the article is well-written and the significant number of recent and relevant references are provided.

Since Table 1 represents important contribution in collecting available data about ciliary genes and proteins involved in cortical malformations, I have several comments and suggestions related to this part:

-        Line 254 (Table description): The colors described as „human (blue) and mouse (orange)“ are not visible in the pdf version of this work. Grey and green columns appear instead. Please solve this issue.

-        Different font size is used in column that refers to Mouse phenotype – it would be more consistent to have the same font size&type everywhere (according to the Journal's instructions for authors, for instance reduced font size would allow a better view).

-        Reference column: it would be better to have numbered references (as in the main text). This would facilitate the reader in accessing specific references of interest. For instance, Ou et al., 2002 becomes ref. no. 55; Wang et al 2009 bocomes no. 51 etc; Dauber et al. 2012 – this article is missing in the references.

-        Why some genes are reported toghether (e.g. CEP35 and STIL or KIF 11,14…) while all the others are described individually?

-        Is there any specific criteria/logic behind the order of the genes/proteins? Are they listed according to their relevance or randomly?

Minor comments:

Formatting errors (different font type and size respect to the rest of the text): lines 331-333, 336, 406-407, 414-415, 417, 434-435, 541.

Line 7 (Abstract): Apical radial glia (aRGs) – „glia“ is already plural form so the correct abbreviation in this case should be „aRG“ (even if the most used abbreviation in this work is „aRGs“ as defined in the line 27 of the Introduction)

Line 18: „Primary“ - not bold

Line 44: „alterations“, „misfunctions“ or similar terms (instead of „ PC problems“)

Line 84: the dot is missing at the end of the sentence

Line 87: „antenna-like“ (instead of „antenna like“)

Line 149: „Taxol-induced“ (istead of „Taxol induced“)

Line 218: replace „cKO“ with „conditional knockout (cKO)“ - first mention

Line 338: In vitro (all italics, not just „vitro“)

Line 342: remove the definition of cKO (abbreviation already defined on page 7)

Line 390: Explain who are „they“?

Figure 3B: Define the meaning of „PM“ (bottom left corner) in the figure legend

Line 408: please specify that „decreased IPs and increased neurons” refers to their number (and not for instance to their size)

Line 460: use abbreviation “CR” for ciliary remnant (defined previously at line 200)

Line 494: “substrate-based” (instead of substrate based)

Line 570: “…” unnecessary
Line 612: other structures (instead of “etc”)

References:

Ref. no.6 (Agirman et al, 2017) is numbered as “1”

Ref. no. 19 and 30 – the authors & journal are missing

Section numbering:

Line 535: replace „2.3.1.“ with „3.3.1“ and the same for the line 554 („3.3.2“ instead of „2.3.2“)

English language:

Some sentences are not very clear and/or are quite long such as:

Line 105 (Figure 3 legend): „helps block simple diffusion“ -check grammar

Line 128: Split the sentence in two:  „…PC formation. On the other hand, …”

Line 166-7: sentence too long („cilia formation at times“ -not clear)

Line 185-186: „For the G1/S transition, the role of trichoplein can be noted, this activates Aurora A to inhibit cilia formation and triggers deciliation“.  (it would be more clear to say: „trichoplein activates Aurora A“)

Line 194: „additional proteins“ (instead of „further“)

Line 455-456: „Notably, centrosomes are conserved by the detaching cells with excision of cilia to remain attached to abscised apical membrane.“- unclear sentence

Author Response

In this review the authors provide an extensive overview of the role of primary cilia (PC), protrusions that form on the apical process of radial glia during early corticogenesis. In addition to centrosomes, they are associated with many important functions of apical radial glia allowing the proper cortical development to occur. The author provide a detailed description of the PC formation, structure and functions as well as a "catalogue" of ciliary genes/proteins linked to the brain malformations when mutated (listed in Table 1 with brief description of their function, both in wild type and mutated phenotype and with references for the specific studies performed). The rest of the article is well-written and the significant number of recent and relevant references are provided.

We thank this reviewer for taking the time to read carefully our manuscript and for the pertinent observations and comments given. We are pleased that our work was appreciated and mentioned as useful. We have taken in account all the recommendations to improve the manuscript and especially Table, including the indicated formatting and grammar corrections. We hope the reviewer will find all information suitably improved.

Since Table 1 represents important contribution in collecting available data about ciliary genes and proteins involved in cortical malformations, I have several comments and suggestions related to this part:

-        Line 254 (Table description): The colors described as „human (blue) and mouse (orange)“ are not visible in the pdf version of this work. Grey and green columns appear instead. Please solve this issue.

We apologize for this problem which we were initially unaware of. We have changed the description to grey and green.

-        Different font size is used in column that refers to Mouse phenotype – it would be more consistent to have the same font size&type everywhere (according to the Journal's instructions for authors, for instance reduced font size would allow a better view).

Once again, we apologize for this problem. We have completely re-done the Table to avoid such problems and we hope this problem does not re-appear.

-        Reference column: it would be better to have numbered references (as in the main text). This would facilitate the reader in accessing specific references of interest. For instance, Ou et al., 2002 becomes ref. no. 55; Wang et al 2009 bocomes no. 51 etc;

We edited the table to now cite references in numbered form.

Dauber et al. 2012 – this article is missing in the references.

Thank you for this observation, we have now added this reference in our new reference list.   

-        Why some genes are reported toghether (e.g. CEP35 and STIL or KIF 11,14…) while all the others are described individually?

We initially had the idea of grouping functionally certain categories of genes. We agree it is much clearer to separate these genes and we have now done this in the new version of the Table.

-        Is there any specific criteria/logic behind the order of the genes/proteins? Are they listed according to their relevance or randomly?

Originally we favorised listing genes involved in human malformations above and mouse mutants below in our Table. However, we agree this became less obvious as more information was added and the list began to appear less well-ordered. We have now changed the order of the proteins and genes to appear more reasonably in an alphabetic order. We believe this should help future interrogation of the Table by readers.

Minor comments: (all now rectified in the new version of the manuscript to our knowledge)

Formatting errors (different font type and size respect to the rest of the text): lines 331-333, 336, 406-407, 414-415, 417, 434-435, 541.

Line 7 (Abstract): Apical radial glia (aRGs) – „glia“ is already plural form so the correct abbreviation in this case should be „aRG“ (even if the most used abbreviation in this work is „aRGs“ as defined in the line 27 of the Introduction)

Line 18: „Primary“ - not bold

Line 44: „alterations“, „misfunctions“ or similar terms (instead of „ PC problems“)

Line 84: the dot is missing at the end of the sentence

Line 87: „antenna-like“ (instead of „antenna like“)

Line 149: „Taxol-induced“ (istead of „Taxol induced“)

Line 218: replace „cKO“ with „conditional knockout (cKO)“ - first mention

Line 338: In vitro (all italics, not just „vitro“)

Line 342: remove the definition of cKO (abbreviation already defined on page 7)

Line 390: Explain who are „they“? This sentence has now been removed.

Figure 3B: Define the meaning of „PM“ (bottom left corner) in the figure legend

Line 408: please specify that „decreased IPs and increased neurons” refers to their number (and not for instance to their size)

Line 460: use abbreviation “CR” for ciliary remnant (defined previously at line 200)

Line 494: “substrate-based” (instead of substrate based)

Line 570: “…” unnecessary
Line 612: other structures (instead of “etc”)

We edited the manuscript taking into account all these minor comments of the reviewer.

References:

Ref. no.6 (Agirman et al, 2017) is numbered as “1”

Ref. no. 19 and 30 – the authors & journal are missing

We have now re-done the reference list taking into account these identified problems.

Section numbering:

Line 535: replace „2.3.1.“ with „3.3.1“ and the same for the line 554 („3.3.2“ instead of „2.3.2“)

We edited as suggested and thank the reviwer for noticing this “typo“.

English language:

Some sentences are not very clear and/or are quite long such as:

Line 105 (Figure 3 legend): „helps block simple diffusion“ -check grammar

We have now improved this.

Line 128: Split the sentence in two:  „…PC formation. On the other hand, …”

We have now improved this.

Line 166-7: sentence too long („cilia formation at times“ -not clear)

We have now improved this.

Line 185-186: „For the G1/S transition, the role of trichoplein can be noted, this activates Aurora A to inhibit cilia formation and triggers deciliation“.  (it would be more clear to say: „trichoplein activates Aurora A“)

Thank you, this is now improved as suggested.

Line 194: „additional proteins“ (instead of „further“)

This modification was performed as suggested.

Line 455-456: „Notably, centrosomes are conserved by the detaching cells with excision of cilia to remain attached to abscised apical membrane.“- unclear sentence

We agree this was not clear and the sentence has now been improved.

Throughout the manuscript we tried to simplify incoherent, long and confusing sentences. We hope now that we have been able to better express our ideas and provide simplified reading of this review.

Round 2

Reviewer 1 Report

The revised manuscript has improved significantly. I am satisfied with the revision.